# Transcriptome and Metabolome Analyses Provide Insights into the Watercore Disorder on “Akibae” Pear Fruit

**DOI:** 10.3390/ijms22094911

**Published:** 2021-05-06

**Authors:** Xiao Liu, Hui-Ming Fan, Dong-He Liu, Jing Liu, Yan Shen, Jing Zhang, Jun Wei, Chun-Lei Wang

**Affiliations:** School of Horticulture and Plant Protection, International Research Laboratory of Agriculture and Agri-Product Safety, Key Laboratory of Plant Functional Genomics of the Ministry of Education, Yangzhou University, 48 Wenhui East Road, Yangzhou 225009, China; liuxiao@yzu.edu.cn (X.L.); fanhuiming2021@163.com (H.-M.F.); ldh1402@163.com (D.-H.L.); liujing1026044@163.com (J.L.); shenyan950148@126.com (Y.S.); zhangj45@yzu.edu.cn (J.Z.); weijun@yzu.edu.cn (J.W.)

**Keywords:** sand pear, watercore, transcriptome, metabolome

## Abstract

Watercore is a physiological disorder that commonly occurs in sand pear cultivars. The typical symptom of watercore tissue is transparency, and it is often accompanied by browning, breakdown and a bitter taste during fruit ripening. To better understand the molecular mechanisms of watercore affecting fruit quality, this study performed transcriptome and metabolome analyses on watercore pulp from “Akibae” fruit 125 days after flowering. The present study found that the “Akibae” pear watercore pulp contained higher sorbitol and sucrose than healthy fruit. Moreover, the structure of the cell wall was destroyed, and the content of pectin, cellulose and hemicellulose was significantly decreased. In addition, the content of ethanol and acetaldehyde was significantly increased, and the content of polyphenol was significantly decreased. Watercore induced up-regulated expression levels of sorbitol synthesis-related (sorbitol-6-phosphate dehydrogenase, S6PDH) and sucrose synthesis-related genes (sucrose synthesis, SS), whereas it inhibited the expression of sorbitol decomposition-related genes (sorbitol dehydrogenase, SDH) and sorbitol transport genes (sorbitol transporter, SOT). Watercore also strongly induced increased expression levels of cell wall-degrading enzymes (polygalactosidase, PG; ellulase, CX; pectin methylesterase, PME), as well as ethanol synthesis-related (alcohol dehydrogenase, ADH), acetaldehyde synthesis-related (pyruvate decarboxylase, PDC) and polyphenol decomposition-related genes (polyphenol oxidase, PPO). Moreover, the genes that are involved in ethylene (1-aminocyclopropane- 1-carboxylate oxidase, ACO; 1-aminocyclopropane- 1-carboxylate synthase, ACS) and abscisic acid (short-chain alcohol dehydrogenase, SDR; aldehyde oxidase, AAO) synthesis were significantly up-regulated. In addition, the bitter tasting amino acids, alkaloids and polyphenols were significantly increased in watercore tissue. Above all, these findings suggested that the metabolic disorder of sorbitol and sucrose can lead to an increase in plant hormones (abscisic acid and ethylene) and anaerobic respiration, resulting in aggravated fruit rot and the formation of bitter substances.

## 1. Introduction

Watercore is an internal physiological disorder affecting apples and pears, in which the intercellular air spaces of the flesh become filled with liquid, resulting in tissues with a translucent appearance [1]. Although the watercore of apples is sometimes desired as an indicator of full ripeness and it has been described as having a sweet or sweetish fermented flavor, in some pear cultivars the affected area is slightly brown and tastes bitter, making the watercore pears not acceptable for consumption [2].

Watercore can be found most frequently in late-maturing fruit. A large number of studies have examined environmental, physiological and biochemical factors involved in watercore development [3]. So far, the solution has not been identified precisely. In recent years, most studies have suggested that the accumulation of sorbitol in the intercellular spaces is a possible cause of watercore in apples [4]. A wide range of environmental and physiological factors have been implicated in the incidence of watercore. In general, there are three main factors contributing to watercore incidence: (1) Abnormal temperature. For instance, watercore fruit in the “Hosui” and “Niitaka” pear species was found to increase significantly under higher air temperatures during fruit maturation [5]. (2) Plant hormones. In pear production, gibberellin (GA) is widely used to increase fruit size and promote ripening. However, the exogenous application of GA to “Hosui” fruit resulted in an increase in fruit size and watercore incidence [6,7]. On the contrary, the GA inhibitor paclobutrazol can suppress watercore formation in pears [8]. (3) Mineral nutrition deficiency. Low fruit calcium is supposed to be an important cause of watercore in apples. Moreover, watercore can accelerate the softening of fruit and make it not suitable for storage [3]. Yamaki and Kajiura reported that watercore tissues have a lower hemicellulose and cellulose content than non-watercored tissues [9]. In addition, watercored tissues show increased activity of endocellulase, polygalacturonase, β-galactosidase, xylanase, and arabinase, which participate in pectin, hemicellulose and cellulose production [10]. Even though there are many studies on watercore, most of them stay at the level of physiological research, and the in-depth molecular mechanism is rarely reported.

In Asia, apples with watercore are preferred and considered a delicacy because of their enhanced sweet flavor. However, the significant difference in the sweetness of a watercore apple is likely affected by components other than sugars. Lee and Lumpkin considered these profiles of flavor components and sensory attributes, and determined that the contribution of aroma components, such as ethyl esters, is crucial in producing the flavor characteristics in watercore-rich apples. They appear to be potent, key flavor compounds in watercored apples [11,12]. However, watercore pear not only has no sweet aroma but also smells of alcohol and tastes bitter. The changes of metabolism related to taste have not been studied in pear.

There are large cultivar differences among apples and pears in the susceptibility to watercore. Compared to the number of resistant cultivars, there are a large number that are susceptible. “Akibae” pear is a late-maturing variety, and it has high productivity, high soluble solid content and good storage potential. “Akibae” pear is a self-compatible cultivar and is resistant to black spot disease. However, it is highly susceptible to watercore [13]. Recent technical advancements in transcriptome and metabolome analyses have provided effective ways to identify new genes and metabolites and to elucidate complex secondary metabolic bioprocesses in plants. To date, there have been few high-throughput omics studies on pear watercore. In pears, only Nishitani et al., using microarray methods, identified 115 differentially expressed genes preceding watercore development [1]. Here, the internal changes of the molecular mechanisms of “Akibae” fruit were analyzed using RNA-seq and LC-MS/MS to understand the changes of gene expression and metabolites during the deterioration of the fruit’s internal quality, which might allow an understanding of the preventive measures to fruit watercore.

## 2. Results

### 2.1. RNA Sequencing and Analysis of Differentially Expressed Genes

At 125DAF, the “Akibae” fruit displayed severe watercore symptoms; the pulp became transparent and water-soaked (Figure 1A). The transcriptome changes of watercore pulp were investigated through RNA-Seq analysis (Table 1). More than 40 million reads were generated per sample. Of these reads, the Q30 percentage (sequencing error rate < 1%) was over 93%, and GC content was approximately 47% for the libraries. Among all the libraries, 78.31–79.80% of unique reads were mapped to the *Pyrus betulifolia* genome. A total of 5218 genes and 1732 genes in watercore pulp were significantly up-regulated and down-regulated, respectively (Figure 1B).

All the DEGs were subjected to GO and KEGG analysis. In the GO database, translation peptide and biosynthetic processes were the most enriched in the “biological process” category; ribosome, ribonucleoprotein complex, non-membrane-bounded organelle and intracellular non-membrane-bounded organelle were the most enriched in the “biological process” category cellular component; structural constituents of the ribosome, structural molecule activity, ADP binding, oxidoreductase activity, acting on the CH–OH group of donors and calcium ion binding were the most enriched in the “molecular function” category (Figure 1C). Pathways showing significant change (Q value ≤ 0.05) in watercore pulp were identified using the KEGG database. The DEGs were involved in seven enriched pathways: ribosome, protein processing in the endoplasmic reticulum, carbon metabolism, plant hormone signal transduction, biosynthesis of amino acids and spliceosome (Figure 1D).

To confirm their authenticity, ten DEGs were randomly selected to analyze their expression profiles by qRT-PCR. The results of qRT-PCR analysis showed that the expression profiles of DEGs were similar to those obtained through high-throughput sequencing (Appendix A).

### 2.2. Changes of Sugar Components Content and Their Metabolism-Related Gene Expression in Watercore Fruit

Sugar content analysis showed that the sorbitol and sucrose were significantly higher in watercore pulp when compared with healthy pulp (Figure 2A). The analysis of DEGs found that two sucrose synthase genes (Chr11.g13337 and Chr2.g41898) were significantly up-regulated (Figure 2A). Moreover, a sorbitol synthesis gene, S6PDH (Chr10.g16781), was significantly up-regulated (Figure 2B). The expression levels of two sorbitol dehydrogenase genes (Chr1.g56757 and Chr5.g08993) and four sorbitol transporter genes (Chr2.g41504, Chr10.g16536, Chr5.g08607 and Chr5.g08618) were significantly decreased in watercore pulp (Figure 2B,C).

### 2.3. Transmission Electron Microscopy (TEM), Cell Wall Components Analysis and Their Metabolism-Related Gene Expression in Watercore Fruit

The watercore pulp cells were seriously damaged, the cell wall was decomposed and seriously separated and only a small amount of cellulose and cell wall remained (Figure 3A). Quantitative analysis showed that the content of pectin, cellulose and hemicellulose in watercore pulp was decreased 55.85%, 41.76% and 57.48%, respectively (Figure 3B). Several DEGs that are involved in cell wall degradation were identified in watercore fruit, including five CX genes (Chr16.g31276, Chr8.g55557, Chr8.g55553, Chr11.g13444, Chr6.g51899), eight PME genes (Chr7.g34150, Chr8.g55784, Chr11.g13340, Chr1.g56500, Chr11.g13339, Chr15.g04724, Chr3.g17608, Chr1.g56500) and seven PG genes (Chr11.g13398, Chr7.g31814, Chr1.g56631, Chr10.g15220, Chr16.g30583, Chr12.g36266, Chr3.g18628), and these were significantly increased (Figure 3C).

### 2.4. Analysis of Ethanol, Acetaldehyde and Total Phenols Content and Their Metabolism-Related Gene Expression in Watercore Fruit

As shown in Figure 4A, the ethanol, acetaldehyde and total phenols contents in watercore pulp were significantly increased. The results showed that the ethanol and acetaldehyde content increased 710.26% and 100.54%, respectively. Total phenol content was 0.25 mg·g^-1^ in healthy fruit, whereas it was not detected in watercore fruit. The expression levels of six alcohol dehydrogenase genes (Chr2.g42515, Chr5.g09476, Chr5.g09474, Chr10.g17249, Chr5.g07504, Chr1.g56327), three pyruvate decarboxylase genes (Chr10.g14289, Chr4.g40093, Chr12.g35311) and two polyphenol oxidase genes (Chr10.g14116, Chr10.g14114) were significantly increased in watercore fruit (Figure 4B).

### 2.5. Analysis of Ethylene and Abscisic Acid Metabolism-Related Gene Expression in Watercore Fruit

The expression levels of ethylene synthesis-related genes, including three ACS (Chr14.g49127, Chr6.g52270, Chr1.g57923) and ten ACO genes (Chr9.g46362, Chr10.g15143, Chr17.g25651, Chr15.g03051, Chr5.g05709, Chr17.g25647, Chr13.g22435, Chr17.g25654, Chr12.g37744, Chr13.g22438), were significantly increased in watercore fruit (Figure 5A). Moreover, six SDR genes (Chr1.g58115, Chr1.g58111, Chr1.g58116, Chr4.g40158, Chr7.g33383, Chr6.g52431) and six AAO (Chr8.g53698, Chr11.g11731, Chr11.g11716, Chr8.g53700, Chr15.g04799, Chr8.g54319) genes that are involved in ABA synthesis were significantly up-regulated (Figure 5B).

### 2.6. Metabonomic Analysis and Identification of Bitter Taste Substances in Watercore Fruit

LC-MS untargeted metabolomics techniques identified 463 differential metabolites, of which 275 were significantly increased and 188 were significantly decreased, respectively. As shown in Appendix A, the annotated differentially expressed metabolites (DEMs) in watercore pulp compared to healthy pulp of “Akibae” were mainly enriched in the biosynthesis of amino acids, tryptophan metabolism, biosynthesis of unsaturated fatty acids and phenylpropanoid biosynthesis. In addition, the bitter-tasting substances in plants are mainly divided into polyphenol, alkaloids and amino acids. The major increased amino acids were histidine, arginine, leucine, isoleucine, phenylalanine, tryptophan and lysine. Three alkaloids’ contents were increased, guanosine monophosphate, xanthine and indole. Neohesperidin dihydrochalcone and naringin dihydrochalcone, which belong to flavonoids, were significantly increased (Table 2).

## 3. Discussion

In apple, the excessive accumulation of sorbitol is considered to be the direct cause of the watercore symptom, and it is closely related to the influence of environmental factors, such as temperature. In Japan, late watercore was shown to be more severe in colder regions among fruit harvested at a similar stage of maturity. Yamada et al. investigated the temperature effect on watercore development in “Fuji” apples by controlling fruit temperature. They found that temperatures of about 25 °C completely inhibited watercore, whereas lower temperatures (10 °C) resulted in a high incidence [14]. However, the effect of temperature on the watercore in pear is the opposite. Hayama found that the watercore disorder in the Japanese pear “Niitaka” is increased by high fruit temperatures during fruit maturation [5]. Liu also found, in “Wonhwang” pear fruit flesh, that the accumulation of sorbitol and sucrose was increased at the time of harvest in high temperatures. Furthermore, SDH enzyme activity and the expression of *PpSDH1* and *PpSDH2* were decreased [15]. In this study, we found that “Akibae” watercore fruit pulp had higher sorbitol and sucrose contents than healthy pulp. The expressions of one S6PDH gene and two SS genes were significantly increased; the expressions of three SDH and two SOT genes were significantly decreased. This suggests that the excessive accumulation of sorbitol and sucrose was a causative factor in the development of late watercore. However, Chun reported that sorbitol metabolism was unlikely to be the primary cause of watercore development in “Akibae” that were planted in Tottori, Japan, because sorbitol did not increase much in “Akibae” and the sucrose content was significantly increased, but “Housui” watercore fruit accumulated higher sorbitol and little sucrose [7]. In summary, the occurrence of watercore is closely related to the disorder of sugar components metabolism, but the inducement of watercore in apples and pears is quite different. Even different varieties of pear will have great differences. The main changes in sugar components may be closely related to the temperature. In China, sand pear is mainly cultivated south of the Yangtze River, where the environment is mild and humid, but the temperature is too high during the fruit ripening period (the average temperature is more than 30 °C). This may be the reason why the “Akibae” watercore fruit had higher sorbitol contents in this study, which may not be the case in Japanese orchards that are located at higher latitudes.

In apples, water-soaked areas are usually found near the core and around the primary vascular bundles, and this has little effect on fruit firmness [16]. However, most pear watercore symptoms occur mainly in the flesh near the skin and seriously affect the fruit firmness, and thus the watercore fruit is not resistant to storage and transportation. The cell walls of apple and pear fruit consist mainly of cellulose, hemicellulose and pectin. Fruit softening during ripening is a complex process that occurs as a result of the expression of a number of hydrolase and trans-glycosylase genes. Among these genes, an increase in enzyme activity and mRNA levels reveals the role of polygalacturonase, cellulase and pectin methylesterase during fruit ripening [17]. In this study, watercore increased the expression of cell wall-degrading enzymes, and the content of cell wall components was significantly decreased. Similar results have been reported [9]. However, it is unclear whether these changes in the cell wall components of the affected tissues were directly induced by watercore or indirectly by other factors associated with maturity. Yamada found that the metabolism of sorbitol could be active even in watercored apples, and the accumulated sorbitol in the intercellular spaces might be primarily due to active unloading from the phloem and not increased leakage from the cells [4]. This may indicate that the cell structure is relatively complete in the early stages of watercore. Moreover, this study found that the expressions of ethylene and ABA synthesis-related genes were significantly increased. Ethylene and ABA play a major role in regulating the ripening and softening of climacteric fruit and, accordingly, the expression of some ripening-related cell wall-associated genes and activities, including those of PGs, PMEs and CX [18,19,20]. ABA and ethylene are well known for their pivotal roles in stress response. Stress can induce ethylene and ABA biosynthesis [21,22]. Thus, we speculate that the excessive accumulation of sorbitol in the watercore caused fruit stress, and then the excessive sorbitol induced the increase in ethylene and ABA content, leading to the degradation of cell wall components.

Other detriments of watercore fruit include a bitter taste, and the dark color and even browning of fruit in the late-ripening stages. In this study, the content of total phenols was significantly decreased, and the expression of PPO genes was significantly increased in watercore fruit. PPO can catalyze the oxidation of a variety of phenols to quinones and then polymerize them to melanin, which gradually turns black with time [23]. This may be the main reason for the color change of watercore pulp. Moreover, ethanol and acetaldehyde are the products of the anaerobic respiration metabolism of fruits. The content of ethanol and acetaldehyde and the gene expression of synthesis-related enzymes were significantly changed. Sorbitol is a hydrophilic substance that is extensively accumulated in the intercellular space leading to the hypoxic environment of pulp cells, and then it changes the metabolism of other substances. Moreover, Tanaka found that ethyl ester synthesis, which is crucial in the aroma and flavor profiles of apples, is enhanced under hypoxic conditions within watercored tissues, resulting in a distinctive, fermented flavor [24,25]. In this study, the watercore pulp anaerobic respiration, and the content of ethanol and acetaldehyde increased significantly. Most L-type amino acids have a bitter taste, among which phenylalanine and tryptophan have the strongest bitter taste [26]. In addition, all alkaloids have the characteristics of a bitter taste. The representative substances are purine, indole and steroids [27]. Plant polyphenols are the general name of a class of polyhydroxy phenolic compounds. As plant secondary metabolites, they widely exist in vegetables, fruits and other derivatives, and are an important source of a bitter taste. Generally, bitter polyphenols are divided into flavonoids, phenolic acids, coumarins and tannins. According to the different molecular weights, the taste characteristics of phenols are slightly different. Low-molecular weight phenols, such as flavonoid monomers, tend to be bitter [28,29]. Metabolomics analysis showed that the content of bitter taste-related amino acids, alkaloids and flavonoids was significantly increased in the watercore fruit (Table 2). This may explain why watercore pears taste worse than watercore apples.

Above all, we speculated a model in which watercore negatively influences pear fruit quality (Figure 6). The changes in the external environment, such as high temperature, lead to increases in sorbitol and sucrose synthesis gene expression or decreases in the gene expression of sorbitol transport and degradation in pulp cells, resulting in the accumulation of sorbitol and sucrose in the pulp. The excessive levels of sugars in cells and intercellular spaces can increase the synthesis of ABA and ethylene, and accelerate the degradation of the cell wall and the softening of the fruit. In addition, excessive sugars can also lead to an increase in anaerobic respiration, which makes polyphenols oxidized and leads to the browning of pulp. The anaerobic respiration of pulp will also accumulate too many bitter substances, resulting in a poor taste. Hence, this study provided further information for understanding the molecular mechanisms of the different reactions of pear pulp to watercore stress.

## 4. Materials and Methods

### 4.1. Plant Materials

Pear fruits (*P. pyrifolia* cv. Akibae) were obtained from an orchard in Yizheng, Jiangsu, China. The “Akibae” were grafted on *Pyrus calleryana,* and the grafted seedlings were planted in a soil environment with a pH of 6.5 to 7.0. A total of three 8-year-old trees of similar size that received uniform sunlight were selected for the experiment. Twelve fruit samples were collected 125 days after flowering (DAF). Pulp tissues were dissected and divided into healthy fruit and watercore fruit. Some fresh pulp was used for transmission electron microscopy assay. The remaining samples were immediately frozen in liquid nitrogen and stored at −80 °C for further use.

### 4.2. Extraction and Measurement of Sugar Component

Sugar extraction and concentration measurements were performed following the protocols of Wang and Miao with some modifications [30,31]. A total of 1 g of frozen powder was extracted with 80% ethanol at 37 °C for 30 min. The sugar solution was collected by 10 min centrifugation at 10,000× *g*, and the residues were re-extracted twice. The residues were redissolved in 1 mL distilled water and deionized on coupled columns of Dowex 50 × 8 and Dowex 1 × 8. The column was eluted with 20 mL distilled water, and the solution was dried and resuspended in 1 mL distilled water for HPLC analysis. The soluble sugars were separated on a Waters Sugar Pak I column at 80 °C using water as the mobile phase with a flow rate of 0.5 mL·min^−1^. The concentration of each sample was calculated by the comparison of known concentrations of sorbitol, fructose, glucose and sucrose.

### 4.3. Transmission Electron Microscopy

The watercore tissue was cut into 1 cm^3^ pieces and fixed in 3% glutaraldehyde (pH 7.4) for 24 h. The sample was flushed with 0.1 M phosphate buffer (pH 7.2) three times and fixed in osmic acid for 2 h. Then the block with acetone dehydration was embedded in Epon-Araldite resin (Ted Pella, 18030, Redding, CA, USA). Semithin sections were made into ultrathin sections, and counterstained with 3% uranyl acetate and 0.3% lead citrate. Then watercore tissue was observed with a JEM-2100F electron microscope [32].

### 4.4. Cell Wall Components, Ethanol, Acetaldehyde and Total Phenols Content Measurements

The ethanol and total phenols contents were determined by the test kit (Beijing Boxbio Science & Technology Co., Ltd., Beijing, China; Suzhou Comin Biotechnology Co., Suzhou, China) according to its manual. There were five replicates per treatment. The acetaldehyde was determined according to the method of Martí by HS-MS system [33]. There were three replicates per treatment.

### 4.5. RNA Extraction and Sequencing

RNA of each sample was isolated using the Trizol kit (Promega, MA, USA) following the manufacturer’s instructions. To gain insight into the transcriptomic dynamics of watercore fruit, RNA-seq analysis was performed using pulp tissues collected from the control fruit and watercore fruit 125 DAF in three biological replicates. For each sample, 1 μg of RNA was used to generate an RNA-seq library with NEB Next^®^ UltraTM RNA Library Prep Kit for Illumina (NEB, Ipswich, MA, USA), and the library quality was assessed on the Agilent Bioanalyzer 2100 system. The libraries were sequenced on the Illumina Novaseq platform by Novagene (Beijing, China). Raw data (raw reads) in FASTQ format were first processed, and clean reads were obtained by removing the reads containing adapter and poly-N and those of low quality. Meanwhile, the Q20, Q30 and GC contents of the clean data were also calculated. All RNA-seq data have been deposited in the National Center for Biotechnology Information (NCBI) Gene Expression Omnibus (GEO) database (GEO accession number: GSE164987). The high-quality clean data were mapped to the reference genome of *Pyrus betulifolia* [34] using Hisat2. Gene expression levels were estimated as fragments per kilobase of transcript sequence per millions (FPKM). Differential expression analysis of two groups was performed using the DESeq2 R package, and *p*-values were adjusted using Benjamini and Hochberg’s approach for controlling the false discovery rate. The differentially expressed genes (DEGs) were screened according to the following criteria: |log_2_^foldchange^| ≥ 1 and corrected *p* < 0.05. The selected DEGs were also annotated with Gene Ontology (GO) and the Kyoto Encyclopedia of Genes and Genomes (KEGG).

### 4.6. cDNA Synthesis and qPCR

First-strand cDNA was synthesized from 1 μg total RNA using the Prime Script^TM^ RT reagent kit with gDNA Eraser (Takara, Tokyo, Japan). Gene-specific primers were designed using the online software Primer3 (http://primer3.ut.ee/) which accessed on 8 October 2020, and the quality and specificity of each primer pair were checked by melting curve analysis and sequencing. Primers are described in Appendix A. qPCR was performed with a Bio-Rad CFX96 instrument (Bio-Rad, Waltham, MA, USA) using ChamQ SYBR qPCR Master Mix (Vazyme Biotechnology, Nanjing, China). The amplification procedures were as follows: a pre-denaturation step of 95 °C for 3 min, 40 cycles of 95 °C for 5 s and 65 °C for 5 s, and a melting curve analysis according to Vazyme Biotechnology’s analysis instructions. The Livak [35] method was employed to calculate gene relative expression levels.

### 4.7. Metabolites Extraction and Analysis

Tissues (100 mg) were individually grounded with liquid nitrogen, and the homogenate was resuspended with prechilled 80% methanol and 0.1% formic acid by a well vortex. The samples were incubated on ice for 5 min and then were centrifuged at 15,000× *g* rpm, 4 °C for 5 min. Some supernatant was diluted to the final concentration containing 53% methanol by LC-MS-grade water. The samples were subsequently transferred to a fresh Eppendorf tube and then were centrifuged at 15,000× *g*, 4 °C for 10 min. Finally, the supernatant was injected into the LC-MS/MS system for analysis. UHPLC-MS/MS analyses were performed using a Vanquish UHPLC system (Thermo Fisher, Waltham, MA, USA) coupled with an Orbitrap Q Exactive^TM^ HF mass spectrometer (Thermo Fisher, Germany) in Novogene Co., Ltd. (Beijing, China). The raw data files generated by UHPLC-MS/MS were processed using the Compound Discoverer 3.1 (CD3.1, Thermo Fisher, Hercules, CA, USA) to perform peak alignment, peak picking and quantitation for each metabolite. The metabolites with VIP > 1, *p*-value < 0.05 and fold change ≥ 2 or FC ≤ 0.5 were considered to be differential metabolites.

### 4.8. Statistical Analysis

The data were evaluated by Duncan’s multiple tests in the ANOVA program of SAS (SAS Institute, Cary, NC, USA). Differences were considered significant at *p* < 0.05. Bar graphs were drawn using the GraphPad Prism 7.0 scientific software (San Diego, CA, USA).

## Figures and Tables

**Figure 1 ijms-22-04911-f001:**
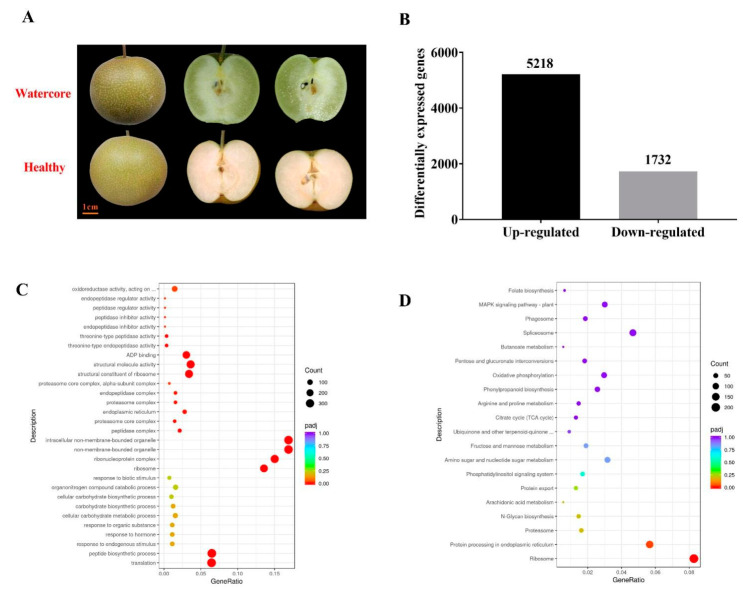
Watercore symptoms, number of DEGs, GO classification and KEGG enrichment of DEGs. (**A**)“Akibae” watercore symptoms; (**B**); Number of DEGs; (**C**) GO classification; (**D**) KEGG enrichment of DEGs.

**Figure 2 ijms-22-04911-f002:**
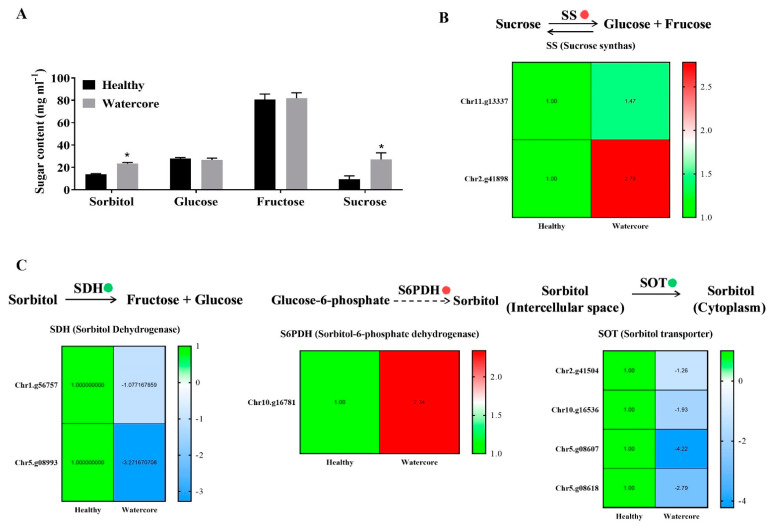
Changes of sugar components’ contents and their metabolism-related gene expression in watercore fruit. (**A**) Changes of sugar components’ contents; (**B**) Changes of sucrose synthas genes expression; (**C**) Changes of sorbitol degradation, synthesis and transport genes expression. The red and green circles refer to up- and down-regulation of gene expression, respectively. ***** The asterisk on the bars indicates significant differences between the watercore and healthy fruit at *p* < 0.05.

**Figure 3 ijms-22-04911-f003:**
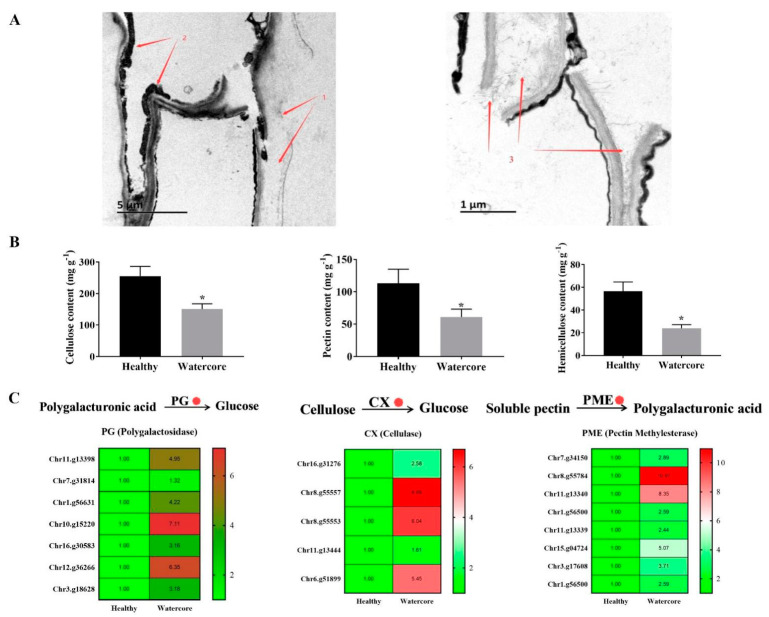
Transmission electron microscopy (**A**), cell wall components (**B**) analysis and their metabolism-related gene expression (**C**) in watercore fruit. Red 1–3 refer to cell wall, residual cytoplasm and cell wall cellulose, respectively. The red circle refers to up-regulation of gene expression. * The asterisk on the bars indicates significant differences between the watercore and healthy fruit at *p* < 0.05.

**Figure 4 ijms-22-04911-f004:**
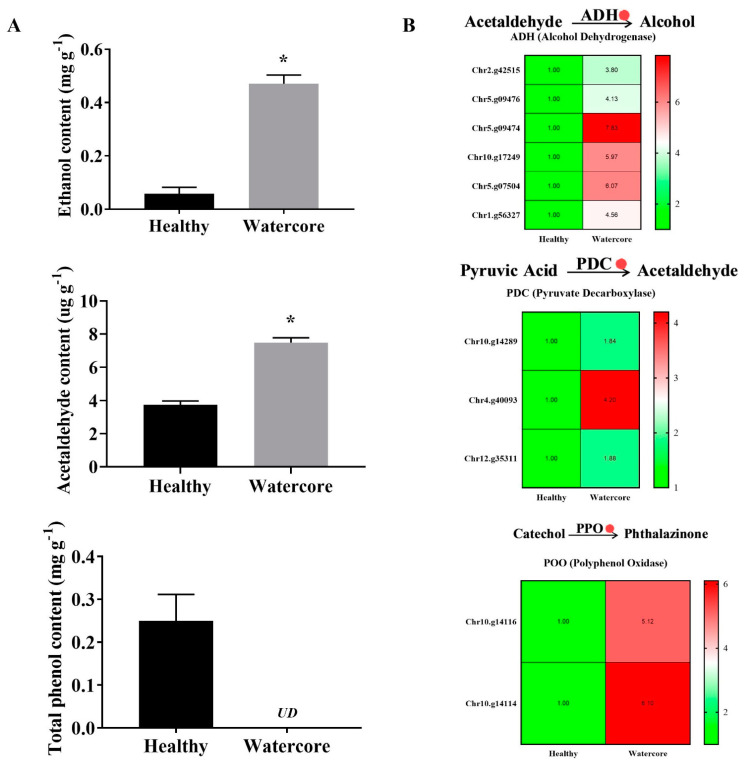
Analysis of ethanol, acetaldehyde and total phenols content (**A**) and their metabolism-related gene expressions (**B**) in watercore fruit. The red circle refers to the up-regulation of gene expression. The asterisk on the bars indicates significant differences between the watercore and healthy fruit at *p* < 0.05. UD: Not detected.

**Figure 5 ijms-22-04911-f005:**
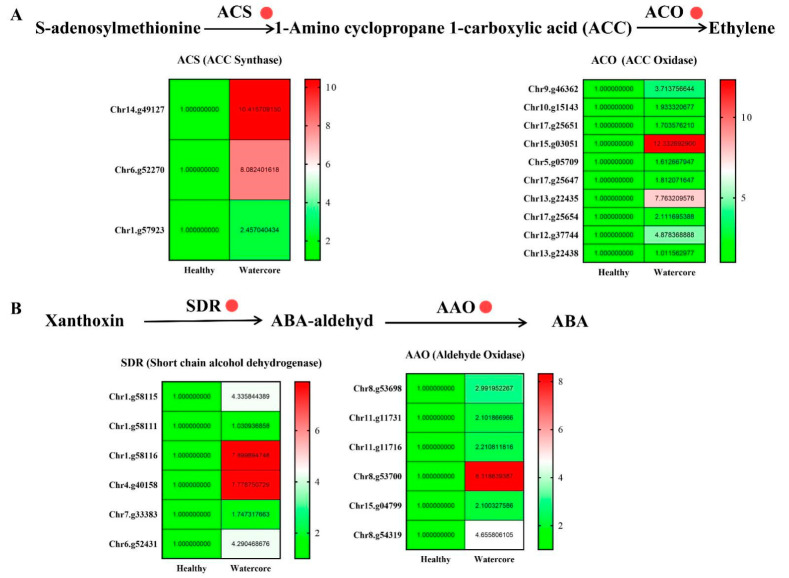
Analysis of ethylene (**A**) and abscisic acid (**B**) metabolism-related gene expression in watercore.

**Figure 6 ijms-22-04911-f006:**
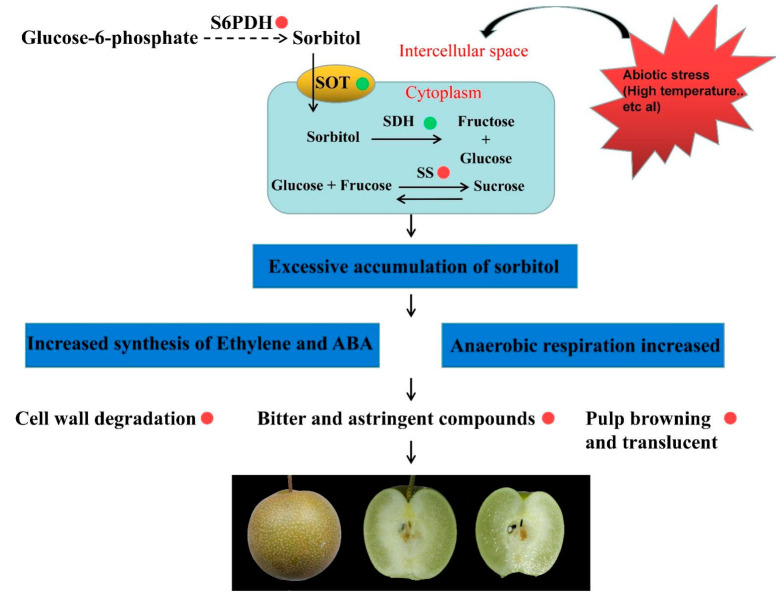
A proposed model of watercore negatively influencing pear fruit quality. The red and green circles refer to promoting and inhibiting related gene expression or metabolic pathways, respectively.

**Table 1 ijms-22-04911-t001:** Statistics of sequencing data of the all libraries.

Statistics of Sequencing Data of the Six Libraries							
Aample	Raw Reads	Clean Reads	Clean Bases	Q20	Q30	GC pct	Total Map
H1	48562128	46972728	7.05G	97.69	93.33	46.78	37245515(79.29%)
H2	45630384	44500836	6.68G	97.92	93.86	46.77	35360796(79.46%)
H3	45673436	44443428	6.67G	97.72	93.4	46.71	35184233(79.17%)
W1	46615168	45091376	6.76G	97.94	93.9	47.71	35310841(78.31%)
W2	47224788	45683826	6.85G	97.84	93.68	47.49	36054606(78.92%)
W3	44767010	43663154	6.55G	97.98	94	47.39	34841297(79.80%)

H1–H3 refer to three healthy fruit replicates; W1–W3 refer to three watercore fruit replicates.

**Table 2 ijms-22-04911-t002:** Different bitter-tasting metabolites in the watercore pulp.

Class of Metabolites	Compound	ID	Log_2_^Fold Change^
Amino acid	Histidine	Com_660_neg	2.73
	Arginine	Com_740_pos;	1.21
		Com_12391_pos	1.61
	Leucine	Com_7242_pos	1.39
	Isoleucine	Com_6_pos	1.06
	Phenylalanine	Com_17_pos	1.06
	Tryptophan	Com_30_pos	3.86
	Ysine	Com_2954_pos	1.28
Alkaloid	Guanosine monophosphate	Com_786_neg	1.24
	Xanthine	Com_4202_neg	2.69
	Indole	Com_500_pos	3.98
Flavonoid	Neohesperidin dihydrochalcone	Com_6076_neg	1.26
	Naringin dihydrochalcone	Com_3268_neg	2.54

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
