# Peer review of "Transcriptome and Metabolome Analyses Provide Insights into the Watercore Disorder on “Akibae” Pear Fruit"

_ijms, 2021, doi:10.3390/ijms22094911_

Round 1
Reviewer 1 Report
The research conducted by the authors is interesting, to understand the watercore mechanism and suggest the factors which can further lead to solving the probelm. The manuscript is well written and easy to follow -however, few minor corrections are suggested below for the overall improvement of the research article.
The comments and suggestions have been appended below -
- Title – Authors wrote sand pear fruit in the title however in the manuscript they are writing it as Akibae pear. I suggest changing the sand pear in the title with Akibae pear will be better—
- Abstract- The format of the journal suggests that the abstract needs to include a small portion of the background, a few lines of methodology and results followed by a conclusion, I suggest authors to follow the guidelines of the journal as in the current format its missing--
- Line no- 66- I think there is some typo error- it should be To date not as to data—
- Citations were not as per the journal's format; I suggest please work on it and include changes in the whole manuscript.
- Results and discussion– No comments and suggestions
- Material and methods- No comments – well described
Author Response
Dear Professor,
We are appreciate the time and effort you have dedicated to providing insightful feedback on ways to strengthen our paper. Thus, it is with great pleasure that we resubmit our article for further consideration. We have incorporated changes in new MS version that reflect the detailed suggestions you have graciously provided.
Sincerely,
Wang Chun-Lei
Reviewer 2 Report
A partly annotated version of the manuscript is provided with indication of necessary modifications.

Author Response
Dear Professor,
We are appreciate the time and effort you have dedicated to providing insightful feedback on ways to strengthen our paper. We have carefully revised the grammar and format of the article, and detailed explained the experimental materials in new MS version. The main purpose of this paper is to analyze the changes of gene expression and metabolites in diseased pulp by transcriptome and metabolome. In addition, we conclude that the cause of the disease is the disorder of sorbitol metabolism, and the related molecular mechanism is under study, but not presented in this paper.
Sincerely,
Wang Chun-Lei
Round 2
Reviewer 2 Report
Did not take into consideration the majority of comments provided in the first revision. Please highlight the revision part.
Author Response
Dear Professor:
We are very sorry for the mistake and obscure. Based on your comments and suggestions, we have made revision to the manuscript. The explanations to the proposal on the manuscript are as Response.
Comments 1: Spelling, grammar and writing errors on line 11, 50, 67, 113, 210, 213, 218.
Response: We are sorry for the mistake. The Figures were sent separately in the revision. We have revised and highlight it.
Comments 2: Line 164-167 move to results section no figures in the discussion
Response: Thank you for your suggestion. We have revised it in new MS version.
Comments 3: Line 187 add literature or erase the sentence
Response: Thank you for your suggestion. We add references to this sentence.
Comments 4: Line 195, this is not based on scientific knowledge erase the speculation of add scientific basis for it.
Response: Thank you for your suggestion. ABA and ethylene is well known for its pivotal roles in stress response. Stress can induced ethylene and ABA biosynthesis. Thus, we speculate that the excessive accumulation of sorbitol in the watercore caused fruit stress, and then the excessive sorbitol induce the increase of ethylene and ABA content, leading to the degradation of cell wall components.
Comments 5: Line 223, wrong direction of the study the cause of the disease or disorder must be elucitated not what is happening in the fruit since this is not helpful at all to manage/solve the problem
Response: Thank you for your suggestion. As we explained in last responses to your comments, the main purpose of this paper is to analyze the changes of gene expression and metabolites in watercore pulp by transcriptome and metabolome. In addition, we conclude that the cause of watercore is the disorder of sorbitol metabolism, and the related molecular mechanism is under study, but not presented in this paper.
Comments 6: Line 232, detailed description of plant materials
Response: Thank you for your suggestion. We have added it to Plant and Materials and highlight it.
Comments 7: Line 262, fruit 120DAF
Response: We are sorry for the mistake. In fact it is 125DAF. In the revision, it was replaced by 125DAF.
Comments 8: Line 282, details on this must be provided
Response: Thank you for your suggestion. We add a reference to this sentence.
I hope that my responses have answered your queries satisfactorily. Please feel free to get back to me should you require any further assistance and I would be glad to assist you again.
Sincerely yours,
Chun-lei Wang
School of Horticulture and Plant Protection Yangzhou University
Round 3
Reviewer 2 Report
The manuscript lacks originality.